
# Digital society social interactions and trust analysis model

Aseem Kumar[1], Arun Malik[1], Isha Batra[1], Naveed Ahmad[2] and Sumaira Johar[3]

[1] School of Computer Science and Engineering, Lovely Professional University, Phagwara, India
[2] College of Computer and Information Sciences, Prince Sultan University, Riyadh, Saudi Arabia
[3] Department of Computer Science, BRAINS Institute Peshawar, Peshawar, Asia, Pakistan

## ABSTRACT

During unprecedented events such as COVID-19, the fabric of society comes under stress and all stakeholders want to increase the predictability of the future and reduce the ongoing uncertainties. In this research, an attempt has been made to model the situation in which the sentiment "trust" is computed so as to map the behaviour of society. However, technically, the purpose of this research is not to determine the "degree of trust in society" as a consequence of some specific emotions or sentiments that the community is experiencing at any particular time. This project is concerned with the construction of a computational model that can assist in improving our understanding of the dynamics of digital societies, particularly when it comes to the attitude referred to as "trust." The digital society trust analysis (D.S.T.A.) model that has been provided is simple to configure and simple to implement. It includes many previous models, such as standing models, Schelling's model of segregation, and tipping points, in order to construct models for understanding the dynamics of a society reeling under the effects of a COVID-19 pandemic, misinformation, fake news, and other sentiments that impact the behaviour of the different groups.

## INTRODUCTION

The emotions and sentiments are expressed as memes, cartoons, videos (*Srivastava & Singh, 2021*; *Tripathi et al., 2019*; *Kapoor, 2022*; *Bader & Obeidat, 2020*), movies, articles, tweets, phrases, songs, images and many myriad ways that society endorses at a given time (*Kim, 2021*; *Nedelea, 2020*). The elements of expression of society are communicated and propagated through multiple mediums of media and languages including mixed languages such as Hinglish. Collectively all these elements become what a postmodern market refer to as "market sentiment" (*He et al., 2022*). A term, which refers to how people feel about investments, stock(s) or markets. When referring to the financial state of the economy, this term is typically used with a negative connotation. However, computer scientists and sociologists are attempting to determine the sentiment of the public through the use of sentiment analysis in a similar and same manner. When a society, a nation, or a significant portion of the population is subjected to unanticipated, unexpected, and inexplicable stimuli that are otherwise incomprehensible, sentiment analysis becomes vital. A high degree of entropy and anxiety (*Albery, Spada & Nikčević, 2021*; *Savolainen et al., 2021*) is

Corresponding author
Naveed Ahmad, nahmed@psu.edu.sa

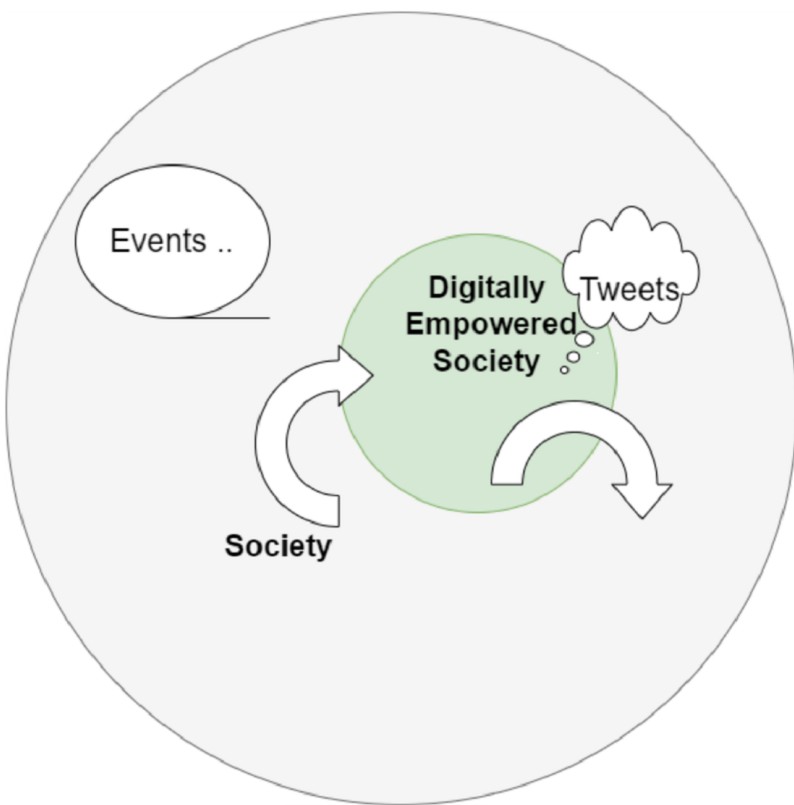

**Figure 1** **Digital societies.**

created in the masses by events such as war, epidemics, famine, and climatic disasters, as well as by any type of socio-political decision that has an impact on a large threshold of population. Today, the speed with which this anxiety spreads is as fast as sending a single tweet post (Fig. 1).

One of the most difficult aspects of dealing with pandemics such as COVID-19 is dealing with the issue of trust. There are rumor mills in operation, pseudoscience are being promoted (*Naeem & Boulos, 2021*; *Ali, Khalid & Zahid, 2021*), and various forms of misinformation and fake news start gaining ground (*Bronstein et al., 2021*; *Bryanov & Vziatysheva, 2021*). They are primarily propagated through digital media and have an impact on the entire social and mental status of society as a whole. There is a pressing need to develop tools that can recreate conditions similar to those that are currently occurring during the COVID-19 era of time (*Altay, Hacquin & Mercier, 2020*; *Flew, 2021*; *Stefanone, Vollmer & Covert, 2019*; *Jaroucheh, Alissa & Buchanan, 2020*). Different communication tactics that help to create trust in society should be compared, and the tools should be able to aid users in appreciating the deep relationships between different types of communication and people's sociological behaviour should be available (*Jaroucheh, Alissa & Buchanan, 2020*; *Ryan, 2021*; *Heuer & Breiter, 2020*; *Rowe & Alexander, 2017*). When developing a model, it is important to emphasize the facts that must be obtained on the influence of communication during future epidemics to develop a comprehensive communication planning tool that will help to eliminate distrust in the community. The

effectiveness of such simulation models can only be successful if the pulse of the society is captured and understood. Research shows that this can be done by conducting sentimental analysis of the text that is publicly available on media platforms such as Twitter, *etc.* Most importantly the analysis must be conducted in the colloquial languages that are spoken and written in specific geo-location.

Models should be parameterized based on the current empirical data collected in the context of the pandemic. Each individual has different characteristics (such as attitude and different levels of social media access) and reacts differently to the same situation, and perceives their environment (such as proximity to the epidemic), reacting differently to different situations. All such parameters must be considered while building the model. The simulation model or mathematical model should have features such as demonstrating two-way communications *ie.*, the influence between digital communication and epidemic and *vice versa*. Finally, it must help the users to build a decision support system that appropriates multiple psychological/social (*Lotito, Zanella & Casari, 2021*) models that help to reduce friction and improve trust in the society. An explorative analysis of this expected model shows that no single equation model or single approach model will be useful in building a comprehensive model (*Lotito, Zanella & Casari, 2021*; *Zhao et al., 2020*). Hence, in the context of the problem undertaken here following models and approaches seems to be relevant as per the contemporary literature survey.

## RELATED WORK

There are numerous techniques that researchers and social scientists equipped with computer algorithms utilize to better comprehend complex themes such as trust, social divisions, behaviour and equilibrium in digital groups, and phenomena such as the spread of diseases in societies.

Contemporary literature frequently cites a model known as Schelling. The Schelling model helps to model society based on racial, income, ethnicity, cults and income segregation criteria. However, the model can also be altered with custom rules, premises or principles to model groups of people. This model generally helps to establish facts about why people segregate or group themselves in a particular way. The degree of segregation or grouping impacts the speed at which a trust travels, especially when the unprecedented events happening such as COVID-19 pandemic. The Granovetter model (*Pollicott & Weiss, 2001*; *Gerhold et al., 2008*) is yet another model that has been extensively used by social scientists to understand the collective behaviour of groups based on specific criteria or some kind of rule. The Granovetter model has been used for understanding the collective behaviour in terms of equilibrium that society has at any given time. Research shows that based on threshold or criteria large groups unintentionally make decisions that overall impact the characteristics of a group/society. In episodes of trust won or lost, people accept unknowingly things they never intended. This alters the future of 'collective trust' invested in specific premises, according to research. People lose or gain confidence and trust with respect to issues in life-based on the evaluation of their peers. The standing ovation model (*Al-Qurishi et al., 2017*) exactly helps us to model this kind of behaviour, especially peer pressure to behave in a certain way. Similar to the standing ovation model,

the 'Identification Model' (*Sivaev, Fang & Mutai, 2020*; *Zhang et al., 2018*) gives a framework to understand the identity based group dynamics. Why do people identify with a cause such as a dog lover, caste, creed or any such kind of identity? In such cases, the flow of 'trust' also depends on the collective behaviour of the group with which the people individually identify themselves. People can associate themselves with concepts such as race, skin color, or education. People also identify themselves with concepts such as anti-vaccination (*American Association for the Advancement of Science, 2013*) or pro-vegetarian. Trust in such cases can be assessed by building identification models (*Sudo, 2017*; *Bielicki, 2021*).

The degree of trust an individual feels ultimately aggregates for the total behaviour of the society. The rational choice theory or the model based on rational Actors Model/ Aggregation Models (*Niazi, Iantovics & Temkin, 2020*; *Cochardt, 2018*) assumes that individual actions lead to aggregate social behaviour. Individually, the theory predicts the agent will choose the action (or consequence) they favour. Percolation models (*Jovanovic & Marek, 2021*), susceptible (SIS) (*Wang, 2020*), and tipping point analysis help us understand the spread of pathogens (*Nova-Reyes, Muñoz-Leiva & Luque-Martínez, 2020*; *Roberts, 1995*; *Anik & Norton, 2020*), Contagiousness and many phenomena such as biological adaptation, technology life cycles etc. The inflection model helps to identify when a phenomenon goes from convex (f''(y) at a given point is greater than zero: f'(x) > 0) to concave or *vice versa*. Trust becomes highly fragile and fragmented in the society and positive communication with masses of the symbols of authority in society needs action to stabilize the dynamics of society. Useful in the context of the current corona-virus kind of situation and its impact on the social fiber.

These events are highly improbable and are rare but when they occur, they bring drastic changes in the dynamics of the society. Some of these events can be predicted in advance and many of the aspects remain unobservable. In such cases, Trust in the digital society can be modelled using the Markov principles. The Markov model attempts to explain a random process that is dependent on the present event but not previous observations (*Sørensen, 1978*; *Bulatetskaya, 2019*; *Winship & Rapoport, 1986*). The Markov Chain Model are generally applied for fully observable dynamic systems and the Hidden Markov model for partially observable systems. Probabilistic or stochastic models seek to predict the behaviour of a randomized independent process. Markov models, on the other hand, attempt to explain non-random processes. Equation-Based models (*Makransky & Lilleholt, 2018*) are based on identified mathematical relationships (input variables and output). Trust can be modelled using mathematical equations by considering trust as a function of other factors.

Since we are interested in using feedback-control principles to design machines that are capable of responding to changes in human trust level in real-time to build and manage trust in the 'human-machine' relationship and 'human to human relationship'. This study gives an illustration of how this mechanism can be modelled.

It must be noted with a good degree of attention that sensationalism sells. Media headlines are designed to make us take subjects such as COVID-19 deaths too seriously. So, the masses pay more attention to what the television reporters are saying. No one

questions or doubts the errors and fallacies in media articles, tweets, or news commentaries. It is a situation where the 'trust' of society takes new manifestations and shapes. Hence, the focus of this study is to study the 'tinkle down effect '(Percolation modelling (*Jovanovic & Marek, 2021*; *Kiss, Miller & Simon, 2017*)) on society in terms of 'trust' in the society. The expression of people in any form (tweets, posts, demonstrations *etc*.) is the stored value of the raw emotions and current sentiment of the masses. Generally, there are two kinds of models that are used by researchers for understanding the spread of epidemics *i.e.*, equation-based models and agent-based models. This section cannot ignore a comparative analysis with respect to the work done here. For better clarity, it must be noted that this research work aims at understanding the changes in the dynamics of the society when the disease spreads and is not the study of 'How disease spreads in the society. This study is entirely focused on raw emotions and sentiments that prevail during such times. If a series of tweets conversation discusses multiple topics related COVID-19 and vaccinations *etc*. The study does not attempt to find '*evidence or no evidence of trust*' in people regarding vaccinations (*Gopichandran, 2017*; *Woko, Siegel & Hornik, 2020*; *Bostrom & Atkinson, 2010*; *Jelnov & Jelnov, 2022*; *Baudier et al., 2021*) but in general how people consider statements and actions as truth worthy for them to follow them (tipping point modelling; *Livina, Martins & Forbes, 2015*). It can be inferred from the summary that no single model or approach can satisfy the current complex situation. The behaviour models are ideally suited to understand COVID-19 and sentiments associated with it. However, multiple simulation modelling approaches such as Monte-Carlo (*Clara-Rahola, 2020*; *McCulloh, Kiernan & Kent, 2020*) are also useful.

In order to work in this domain, the current generation of researchers use a large number of simulators and tools. OpenModelica, OpenSimulator (*Fishwick, 2009*), Opensim, Anylogic (*Holkin & Pavlov, 2021*; *Borshchev, 2014*), and other simulation frameworks are among the most well-known. When it comes to analysing internet-based information, python and the 'R' libraries linked to natural language are heavily utilised (*Zahidi, Younoussi & Al-Amrani, 2021*).

## Digital society trust analysis (D.S.T.A) model

In this section, a discussion on how the sentiment of society can be decoded from the text such as tweets is discussed. At the same time, this section illustrates the steps in the construction of a new qualitative model analysis framework referred to as the Digital Society Trust analysis (D.S.T.A) model. The section further discusses the inverse problem of modelling, or simply the rules and laws that govern the behaviour of the digitally enabled person in terms of trust as a sentiment or emotion that he/she feels as events such as COVID-19 pandemics occur.

**Variables:**

**Assumptions and Axioms:**

(1) It is assumed that at a certain point in time, 't,' the digital society 'p' maintains a balance of observable laws and regulations that keeps neutral perceptions ('pc') and emotions ('tr') with positive values.

**Table  D.S.T.A Simulation parameters.**

| S. No | Variables | Description | Usage |
|---|---|---|---|
| 1 | 'p' | Digital society | Population of the Digital Society |
| 2 | 'r' | Set of dynamic rules | Rules that guide the thinking/behaviour of Individual |
| 3 | 'cpc' | Collective cognitive apparatus' | How to know the world. |
| 4 | Pp | Perception Apparatus | Interface between what we know The world around us |
| 5 | 'ev' | Events Happening in due course of time 't' | Events can be of two types Mediocristan, Extremistan |
| 6 | N | Size of population | Total size of the population |
| 7 | 'pt' | Type of Digital individuals in population 'p' | Three types of populations {indifferent, positive, negative} |
| 8 | 'tr' | Trust | Trust of a digital society member may be positive, negative, or neutral. |
| 9 | 'sdic' | It is a dictionary of emotions. | This variable represents a dictionary of emotions technically representing bag of words of raw emotions that are associated with trust. |

(2) The mental network or cognitive structure of the digital society is steady; there is no mass event/hysteria or some mass degree of psychological shock functioning in the society at this time.

(3) The balance of the digital society shifts when the dynamics of the norms and regulations change as a result of occurrences such as COVID-19. Each individual is impacted, leading to changes in the society as whole.

**Construction of the model:**

Digital society as an 'institution' expresses itself on the internet based on a set of dynamic rules '*r*' that build the society networks and at the same time guide the thinking and behaviour of an individual (Agents) (*Alemanno, 2018*; *García-Magariño et al., 2017*). These set rules are apparent and move over, they in fact are embedded in the cognitive apparatus of each individual **p***i*. Dynamic rules/laws are by definition a cognitive process by which an individual assimilates 'How to interpret the world around it' and at the same time 'How to act in it' or respond to it. It is assumed that these laws/rules are encoded within the individual's cognitive structure. Hence, the full population '*p*' has its own collective apparatus that defines the fabric of the society. The rules/laws at the individual level may not be observable. However, when the actions, objects and events are recorded and are subjected to data mining analysis associated rules start to appear (Fig. 2).

The 'collective cognitive apparatus', 'cpc' (*Rabb et al., 2022*; *Rodič, 2020*) of population '*p*' builds a perception ('pp' = {positive, neutral, negative}). Perception is an interface between what we know and the world around us. The perception has anchored sentiments/emotions such as trust ('tr' = {positive, neutral, negative} which drives the actions of each individual. Perception is, in fact, the application of the rules/laws assimilated in the mental mind. Perception of an individual can be judged with help of Topic Modelling or simply by analysis of 'content/topics/subjects' it responds to on the internet.

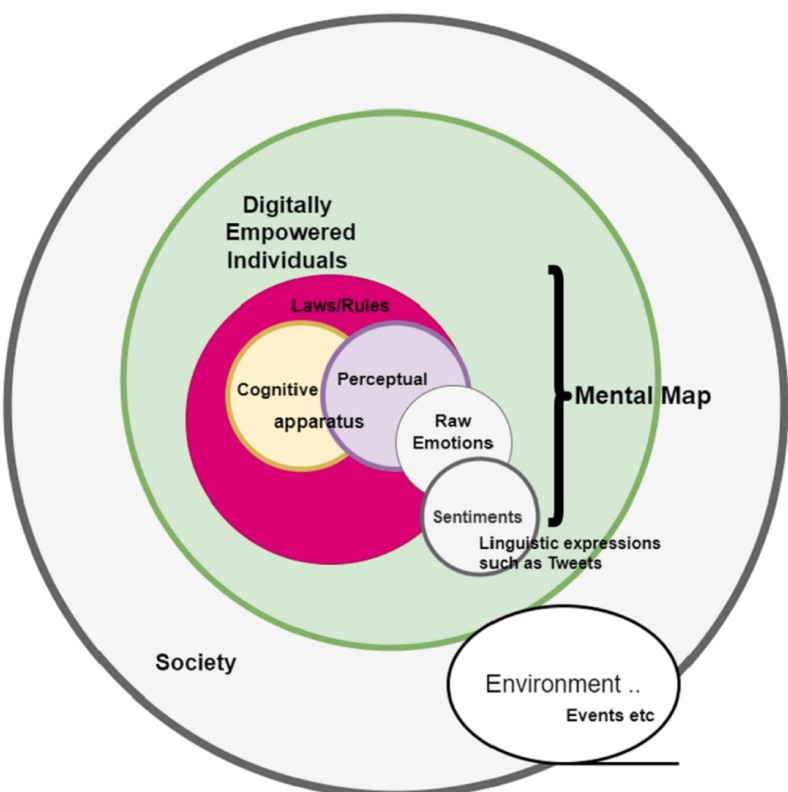

**Figure 2** **Model for understanding sentiments.**

As mentioned in the earlier that at any given time '*t*', the digital society maintains an equilibrium of observable laws/rules that maintains neutral perception '**pc**' and emotion such as trust 'tr = neutral' and at the same time the mental network or cognitive structure of the digital society is stable.

The perceptual apparatus of the individual is a tacit knowledge that guides the thoughts and actions but at the same time it is difficult to linguistically express—it is something we 'just know', but cannot express. This tacit knowledge of the individual about its environmental impacts the emotions to which it is attached such as trust 'tr'.

Emotions (*Ji, Jeong & Jeong, 2019*) are the reasons 'why rules/laws are to be followed and acted upon'. Emotion is not just a state that would be archived with the transformation of sense-data into a motor activity, but it is an axiomatic core of an individual that determines the value of judgment to act upon certain stimuli in the environment. Emotions are linguistically expressible in different forms of words, phrases, tweets, and many other long and short forms of content and principle empirically testable. They have different potentialities, for example, low trust, medium trust or high trust intensities or activations with which the individual may act. Emotions are a kind of *Ipso facto* to give a sense of urgency to act, a kind of motivation to do something in response to the exposed environment. Technically emotion analysis is different from sentiment analysis. Emotions are complex raw states such as fear, happiness etc., while sentiment is an organized form of emotions and is expressed as positive, negative, or neutral. The focus of this research work

is to study the organized state of emotions and perceptions that a digital society has in times of events such as COVID-19. For this research work, it is assumed that linguistic mental maps of individuals can be ground for identifying the potentiality of the sentiment in terms of low, medium and high. The linguistic metal maps of sentiments can be constructed with a *bag of words* (b.o.w) approach. The bag of words represents a specific sentiment dictionary '*sdic*'. With the help of distance metric (distance from a specific linguistic metal map of current sentiment), a 'consequentialist reasoning' can be constructed for knowing the state of sentiment in which the digital society is included.

The equilibrium of the digital society changes as the dynamics of the rules/laws change due to an occurrence of event '**ev**' of type = {Mediocristan, Extremistan (*Thorp & Mizusawa, 2011*; *Ekung, 2014*)}'. In other words, the events bring structural and systematic changes into society. An event may bring a 'multiplicative' impact or no significant impact. The multiplicative impact is a situation when, for example, coronavirus spreads from one person to another at such a speed and it leads to unintended changes in the psyche of society. This changes the previous equilibrium of the society defined or assumed. Modelling methods such as Standing Ovation (*Miller & Page, 2004*) and Segregation models (*Fossett, 2011*) also intend to study such phenomena. However, in this model, we introduce the concept of Mediocristan, Extremistan to understand the sentiments of the digital communities and collectives.

Mediocristan (Fig. 3B) as defined has a thin tail graph when plotted and affects the individual independently of the community. Technically, extremistan affects many individuals. As a result, extremistan brings systemic consequences that mediocristan does not. Multiple events, including diseases, are constantly associated with Extremistan (Fig. 3A). They may be non-lethal (for example, the flu), but they become remnants of Extremistan.

To understand the degree or intensity of a particular sentiment such as trust 'tr' in a population '**p**'. It is critical to understand the nature of the event and the consequences it is generating. The simulation output of the model is given in Fig. 4. The parameters for running the simulation are shown below in Table 1.

When generating the population 'p' of the digital society, the simulation uses default values of 1,000 as the starting point. It is built by applying uniform random distribution to construct the cognitive equipment of the collection of agents (total digital society). It is determined what kind of understanding each individual agent has about his or her socio-economic circumstances by assigning a value to each individual digital agent '$pi$' and a perception '**P**$pi$' of each agent is built. Those who are not oblivious to changes in the socio-economic environment are assigned a higher trust level in this scenario. The population that has chosen to take role in constructing the alterations in the environment is finally chosen for inclusion in the simulation and will be the first to begin. By employing a uniform distribution model ($X \sim U(a = 0, b = 1)$, *where a, b are lower and upper value*), three levels of trust (low, medium, and high) are assigned, and the initial equilibrium is calculated and represented graphically in Fig. 4 showing the zeroth iteration. When the D. S.T.A simulation is running, the society goes through a number of transformations as a

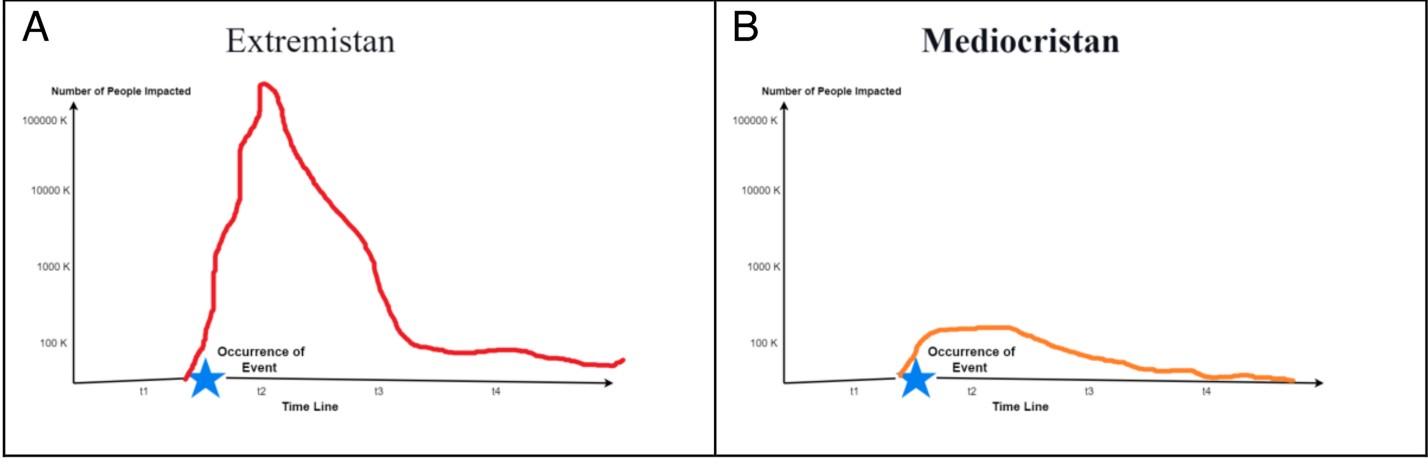

**Figure 3** (A) Extremistan. (B) Mediocristan.

result of events such as COVID-19 no-COVID-19, or some rare type of event that has a significant impact on the society.

The output obtained by the simulator is depicted in Fig. 4. Figure 4 depicts eight different simulated scenarios. The initial trust equilibrium is represented in the first iteration '0', and as the discourse begins because of the advent of some event such as COVID-19 or the Ukraine-Russia war, there is a great deal of motility and emotion runs at a high level. People begin to create new groups or clusters because of the high degree of entropy that has been activated. Individuals and groups begin to change their behaviour in a variety of ways, which leads to polarisation in the society because of the changes. Throughout the simulation, as the various (factors of change) events unfold and trust levels fluctuate, a new equilibrium research is conducted after the conclusion of iteration number 7. If dialogues (tweets) between the digital agents are exchanged regularly (number of conversions exchange threshold = 10), the collective behaviour, attitude, and precipitation in response to environmental stimuli are influenced more. The accumulation of entropy because of events such as COVID-19 causes the trust equilibrium to shift as users begin to influence one another's behaviour, as can be seen in the graph below. For example, the suggested model includes the theory of probability and makes use of structural equations to better comprehend the dynamics of digital society in terms of confidence.

## RESULTS

The innovative computational model given here is an interaction between events and conceptual 'agents' in both space and time, resulting in dynamic patterns and potentially constantly producing behaviour as per the social conditions. Not only do the models allow for variation in agent choices, but also in their behaviour. The agents' of the suggest model's behaviour is "tunable" on multiple dimensions, ranging from hyper-rational in terms of trust sentiment and hyper-informed in terms of how it is empowered to act and interpret happenings such as the COVID-19 epidemic to basic and naive. However, the current work must be evaluated in the backdrop of the Standing Ovation model. The SOM

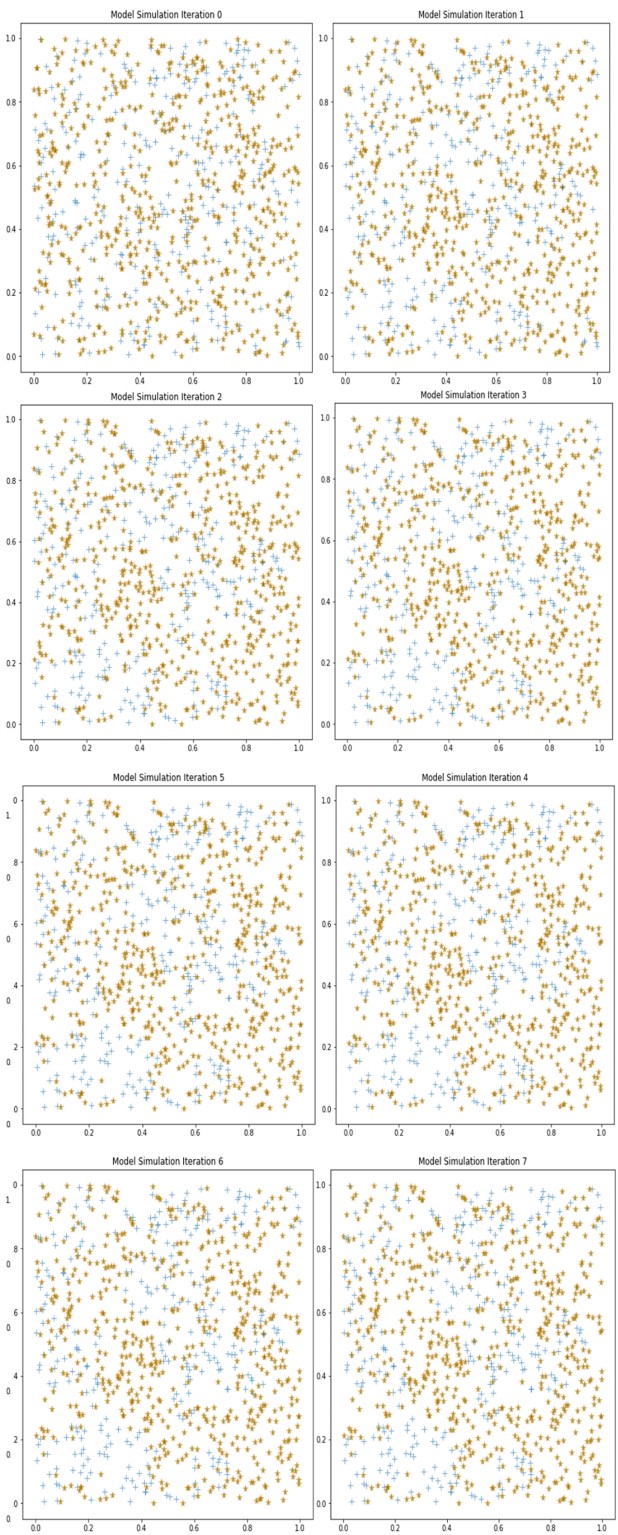

**Figure 4 Movement of trust sentiment in the digital society due to COVID-19.**

**Table 1 Simulation parameters.**

| S. No | Simulation parameter | Value | Mathematical expression |
|---|---|---|---|
| 1 | Population size (Twitter Agents) 'n' | 900 | 'n' |
| 2 | Cognitive apparatus 'c' | {Indifferent, Positive, Negative} | agents = [TwitterAgent for i in range(n)]<br>    c = uniform_randon(1,3) |
| 3 | Perception 'p' | {Positive, Neutral, Negative, Indifferent} | agents = [Cognitive_Apparatus 'c' for i in range(n)]<br>    If! ='Indifferent', c = uniform_random(1,3) |
| 4 | Levels of Trust Identified Using Rule Mining 't' | {Low, Medium, High} | agents = [Cognitive_Apparatus 'c' for i in range(n)]<br>    If! = disinterested,<br>        assess_trust_levels() |
| 5 | Distribution of Trust Levels in Population | Balanced | for each agents<br>    agent_trust_level = rand.uniform(assign_trust_levels()) |
| 6 | Factors of Change Probability | {Mediocristán = [Covid, No Covid], Extremistan = {Rare Event}} | for each agents<br>    update_trust_level<br>        chg_prob = compute_probality_event<br>    (Mediocristán&Extremistan) |
| 7 | Number of Conversions exchange Threshold | 10 | It is the number of messages that have been exchanged with time to influence the behaviour of the other. |
| 8 | Number of Iterations of simulation | 100 | Number of rounds the simulation will run |
| 9 | Termination Condition | | while True:<br>    compute_initial_equilibrium<br>    count += 1<br>    no_change_equilibrium = True<br>    for agent in agents:<br>        old_trust_level = agent.trust_level<br>        agent.ChangeFactor(agents)<br>        if agent.trust_level != old_trust_level:<br>            no_change_equilibrium = False<br>    if no_change_equilibrium:<br>        break |

is a good metaphor for social circumstances in which agents make binary judgments such as sending to private schools or not and interacting locally. In real life, outcomes are usually open-ended, or least, variable. Generally, binary exposures (true or false, pregnant, or not pregnant) are limited in their impact on society hence, the models are limited in this sense. The Granovetter model (*Granovetter, 1973*) has been improvised multiple times with the help of *graph theory* (*Kaveh, 2013*) and *Probability theorems*. The foundation is the computation of the strength of ties and relationships. A concept that is useful in the context of understanding 'social fabric' and trust in society (*Lenton, 2013*). The proposed model incorporates the computations as well related to this. However, the proposed approach allows us to compute the strength of the tie with help of 'tweets or content' expressed by the users. The suggested model involves the detection of tipping points (*Schelling, 1971*), which is a critical component. In addition, it seeks to discover reference points when a small change in the "market sentiment or social sentiment" parameters results in a significant shift in the raw emotion of trust. As a result, the current study

highlights the mechanisms by which information/rumor diffusion happens and the points at which the dynamics of trust alter in situations such as COVID-19.

## CONCLUSIONS AND FUTURE WORK

The principle of the diffusion of information and the principle of social change shows the impact of trust on the behaviour of thousands of people. The role of 'trust' as a sentiment in the diffused network of information highways has been investigated for digitally empowered people.

However, it should be noted that the outcome of such a study depends on a large number of dynamic variables. Hence, this research effort has its limitations, but it is also important to highlight that, in some part, the model's resilience is determined by the person who is looking at it. Because of the ease with which computational modeling may be used, researchers can readily adjust the assumed behaviors and parameters to find the important elements driving the results. At the same time, it can be observed from this proposed model that it has several advantages in terms of flexibility and effectiveness in understanding the social conditions, perceptions and sentiments using 'content expressions' of the people.

The purpose of this research is to lift information above the noise. It tries to blend vivid storytelling and persuasive arguments. By describing experiences with vivid and emotive tales and explanations, you can influence individuals to view events and facts differently. This dynamic has the potential to unite as well as polarise social groups. The model attempts to comprehend this truth within the framework of what may be observed using social media settings. The meaning that individuals ascribe to information varies according to the social groupings to which they belong. Successful information campaigns can be rooted in or targeted towards existing social groups. Leaders and authority figures have a significant influence on how information is interpreted and presented. As a result, this study is also relevant for policymakers who want to uplift the 'trust' in society in a positive manner.

For future studies, it is recommended that extension of this model to include more parameters and events that do not follow extremistan or mediocristan pattern may be done. The current study attempts to deal with many random and uncontrollable variables; the accuracy of such models degrades as time length increases. Hence, there will be a need to redo such studies in light of new circumstances and irregularities that may come up in the future.

### Funding
Prince Sultan University funded the Article Processing Charges (APC) of this publication. The funders had no role in study design, data collection and analysis, decision to publish, or preparation of the manuscript.

## Grant Disclosures

The following grant information was disclosed by the authors:

Prince Sultan University (Article Processing Charges (APC) of this publication).

## Competing Interests

The authors declare that they have no competing interests.

## Author Contributions

- Aseem Kumar conceived and designed the experiments, performed the experiments, analyzed the data, performed the computation work, prepared figures and/or tables, and approved the final draft.
- Arun Malik analyzed the data, prepared figures and/or tables, and approved the final draft.
- Isha Batra analyzed the data, authored or reviewed drafts of the article, and approved the final draft.
- Naveed Ahmad analyzed the data, authored or reviewed drafts of the article, and approved the final draft.
- Sumaira Johar analyzed the data, authored or reviewed drafts of the article, and approved the final draft.

## Data Availability

The simulation code is available in the Supplemental File.

## Supplemental Information

Supplemental information for this article can be found online at http://dx.doi.org/10.7717/peerj-cs.1129#supplemental-information.

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
