# Peer review of "Digital society social interactions and trust analysis model"

_PeerJ Computer Science, doi:10.7717/peerj-cs.1129_

## Round 0.1 · original submission · Major Revisions

Three consistent reviews have been received. A major revision is needed before further processing. Please provide a detailed response letter. Thanks.

Reviewer 1 ·

Basic reporting

The paper has been written clearly, the use of English is quite good, supported by appropriate references and supports the study of the diffusion of information and the principle of social change. Paper is concerned with the construction of a computational model that can assist in improving our understanding of the dynamics of digital societies, particularly when it comes to the attitude referred to as "trust" using the digital society trust analysis (D.S.T.A.) model.

The presentation of figures and tables is good, however, Figure 3 should be explained in more detail, especially to understand the degree or intensity of a particular sentiment.

The paper presents an innovative computational model as an interaction between events and conceptual 'agents' in both space and time, resulting in dynamic patterns and potentially constantly producing behavior as per the social conditions.

These results are supported by simulation studies and studies showing that the mechanisms by which information/rumor diffusion happens and the points at which the dynamics of trust alter in situations such as COVID-19.

Experimental design

The research methodology in this paper includes a theoretical study of how the sentiment of society can be decoded from the text such as tweets supported by a simulation study through the construction of a new qualitative model analysis framework referred to as the Digital Society Trust analysis (D.S.T.A) model with 5 assumptions are given. The fifth assumption, the equilibrium of the digital society changes as the dynamics of the rules/laws change due to an occurrence of an event, needs to be clarified in relation to the concept of Mediocristan, Extremistan as shown in Figure 3.

Validity of the findings

The paper has novelty in the construction of a new qualitative model analysis framework referred to as the Digital Society Trust analysis (D.S.T.A) model. The research stages have been carried out quite well, but still need confirmation of the simulation results in Figure 4 Movement of Trust Sentiment in the digital society due to Covid, especially the reason why the starting point starts with 1,000? And the reason for choosing to apply uniform random distribution to construct the cognitive equipment of the collection of agents (total digital society).

In the simulation results it is stated that If dialogues (Tweets) between the digital agents are exchanged regularly (Number of Conversions exchange Threshold = 10), the collective behavior, attitude, and precipitation in response to environmental stimuli are influenced more. The accumulation of entropy because of events such as covid-19 causes the trust equilibrium to shift as users begin to influence one another's behavior, as can be seen in the graph below. But, there is no picture showing the graph, this needs to be completed.

Additional comments

Paper has novelty in the innovative computational model as an interaction between events and conceptual 'agents' in both space and time, resulting in dynamic patterns and potentially constantly producing behavior as per the social conditions. The simulation results need to be confirmed with the iteration selection assumptions and should be supported by an algorithm for applying the model in the simulation.

Reviewer 2 ·

Basic reporting

In the setting of a worldwide epidemic, this manuscript proposes a digital society trust analysis model to reflect the behavior of society. Several problems need to be considered again.

Experimental design

1. Please check the whole manuscript for grammar errors. Many sentences are not smooth and hard to be understood. Some words are not properly used. Here are some examples. Line 50, “Today, the speed with which this anxiety spreads is....”. Line 217, metal or mental?
2. In Section “Digital Society Trust analysis model”, it is better to provide the specific form of your model whatever in the form of formula or figure.
3. In Subsection “Assumptions, Axioms and Constructs”, please reorganize the content because the current content is not consistent with the introduction in Line 166-171. The structure is confused. Maybe it can be rearranged as three parts: 1) assumption, 2) variable definition, 3) model construction. The first part is to introduce the necessary conditions to make the model reasonable. The second part is to interpret the definition of variables or parameters in the model. The third part is to describe the model in a mathematical form, formal logic form, or other forms and illustrate the difference or advantage of your model compared with others.
4. It is better to put the simulation results of Subsection “Assumptions, Axioms and Constructs” in Section “Results”.
5. Please clarify the contribution or the value of the model. The motivation of the manuscript should be highlighted.

Validity of the findings

No comment.

Reviewer 3 ·

Basic reporting

no comment

Experimental design

The parameters of the model proposed in this paper vary one by the choice of agent and one by the behavior. Regarding the choice of agents, can the heterogeneity of agents (e.g., male or female) be considered ?

Validity of the findings

This paper lacks comparative analysis to highlight the advantages of this paper.

Additional comments

1)The innovation of this paper is not very prominent. Please write down the innovation points in detail.
2) This paper identifies key parameters when small changes in "market sentiment or social sentiment" parameters lead to significant changes in the original sentiment of trust. Please explain rationally what causes these key factors?

---

## Round 0.2 · Minor Revisions

A further revision is needed to address the reviewer's concern. Thanks.

Reviewer 1 ·

Basic reporting

The paper has been revised and now it has a better explanation, especially in Figure 3. The authors have explained the degree or intensity of a particular sentiment

Experimental design

The fifth assumption, the equilibrium of the digital society changes as the dynamics of the rules/laws change due to an occurrence of an event, has been clarified in relation to the concept of Mediocristan, Extremistan as shown in Figure 3.

Validity of the findings

The paper has novelty in the construction of a new qualitative model analysis framework referred to as the Digital Society Trust analysis (D.S.T.A) model. The research stages have been carried out quite well, and it has been explained that the simulation results in Figure 4 Movement of Trust Sentiment in the digital society due to Covid with the reason about the starting point starts with 1,000.

Additional comments

The simulation results have been confirmed with the iteration selection assumptions and supported by an algorithm for applying the model in the simulation.

Reviewer 2 ·

Basic reporting

In the setting of a worldwide epidemic, this manuscript proposes a digital society trust analysis model to reflect the behavior of society. Several problems need to be considered again.

Experimental design

1. Please check the indexes of figures. For example, there is some wrong with Figure 1-3 in the text. Tables should also be numbered. Why the figures are not put in the main text?
2. Please improve the format of the Table ‘Variables’ and complete the description and usage of ‘s’.
3. Please clarify why not to describe the D.S.T.A. model in formulas.

Validity of the findings

1. It is better to implement some comparison experiments to show the advantages of method in this manuscript.

Additional comments

In addition, as said in the response letter, there is no highlighted text in the revised manuscript. In response letter, if the authors response the comments one by one and point out the pages or sections of the revised parts, it would be better and more formal.

---

## Round 0.3 · accepted · Accept

All comments of the reviewers have been addressed. I recommend it for publication.

Reviewer 2 ·

Basic reporting

No comments.

Experimental design

No comments.

Validity of the findings

No comments.

Additional comments

No comments.